# Functional Characterization of LTR12C as Regulators of Germ-Cell-Associated TA-p63 in U87-MG and T98-G In Vitro Models

**DOI:** 10.3390/cells14110852

**Published:** 2025-06-05

**Authors:** Lucia Meola, Sohum Rajesh Shetty, Angelo Peschiaroli, Claudio Sette, Camilla Bernardini

**Affiliations:** 1Department of Neuroscience, Section of Human Anatomy, Catholic University of the Sacred Heart, 00168 Rome, Italy; lucia.meola1@unicatt.it (L.M.); claudio.sette@unicatt.it (C.S.); 2Faculty of Medicine and Surgery, Catholic University of the Sacred Heart, 00168 Rome, Italy; sohumrajesh.shetty01@icatt.it; 3Institute of Translational Pharmacology (IFT), CNR, Via Fosso del Cavaliere 100, 00133 Rome, Italy; angelo.peschiaroli@cnr.it; 4GSTeP-Organoids Research Core Facility, Fondazione Policlinico Universitario Agostino, Gemelli IRCCS, 00168 Rome, Italy

**Keywords:** epigenetics, LTR12C, glioblastoma, GTA-p63, p21

## Abstract

Glioblastoma multiforme (GBM) is a deadly disease known for its genetic heterogeneity. LTR12C is an endogenous retrovirus-derived regulator of pro-apoptotic genes and is normally silenced by epigenetic regulation. In this study, we found that the treatment of two glioblastoma cell lines, T98-G and U87-MG, with DNA methyltransferase (DNMT) and histone deacetylase (HDAC) inhibitors activated LTR12C expression. Combined treatment with these epigenetic drugs exerted a synergistic action on the LTR12C activation in both cell lines, while treatment with each drug as a single agent had a far weaker effect. A strong induction of the expression of the TP63 gene was seen in both cell lines, with the pro-apoptotic isoform GTA-p63 accounting for most of this increase. Coherently, downstream targets of p63, such as p21 and PUMA, were also induced by the combined treatment. Furthermore, we observed a significant reduction in the GBM cell growth and viability following the dual DNMT/HDAC inhibition. These findings reveal that the reactivation of LTR12C expression has the potential to modulate survival pathways in glioblastoma and provide information regarding possible epigenetic mechanisms that can be used to treat this deadly disease.

## 1. Introduction

Glioblastoma multiforme (GBM) is one of the most lethal human cancers, with a 5-year survival rate of <5% [1]. The tumor heterogeneity, aggressiveness, and tendency to recurrence contribute to a poor GBM prognosis. Despite advances in surgery and chemotherapy, therapeutic options remain limited. Surgical resection and chemotherapy with temozolomide (TMZ) have remained the gold standard for the therapy of primary tumors since 2005 [2], despite their scarce efficacy. Thus, more efficacious therapies for GBM are urgently needed. In this regard, immunotherapy is emerging as a powerful anti-cancer approach in some tumor types. Unfortunately, however, GBM has a low mutational burden, resulting in a small amount of neoantigens [3]. In fact, one of the features of glioblastoma is its ability to evade the immune system and, from this perspective regarding the activity of transposable elements (TEs), can contribute not only to genomic instability and mutation accumulation but also to the creation of neoantigens that could be recognized by the immune system. Transposable elements or transposons are DNA sequences with the ability to change their location within the genome. TEs are believed to form nearly half of the human genome and while largely silent, they possess a regulatory function on gene expression [4]. From this perspective, epigenetic regulation to activate the transcription of normally silent TEs in GBM could present different outcomes, such as enhancing the production of neoantigens and triggering a strong apoptotic response.

Transposable elements can be further divided into retrotransposons (Class I), which “jump” throughout the genome via reverse transcription, and DNA transposons (Class II), which do not require an intermediate to do the same. Endogenous retroviruses (ERVs) are a subtype of retrotransposons that are remnants of infectious exogenous retroviruses that became fixed in the human cell DNA and are inherited in a Mendelian manner [4]. Recent studies suggest that epigenetic therapies can synergize with immunotherapies through the de-repression of ERVs [5]. ERVs consist of two long terminal repeats (LTRs) flanking the open reading frames (ORFs) that encode the viral proteins. During evolution, ERVs are often reduced to a single LTR, or “solo LTR”, which renders them incapable of retro-transposition but retain their regulatory capacity [6].

The reactivation of ERVs has broad and often paradoxical repercussions since it may both promote and hinder cancer development. Some pro-tumoral consequences mediated by transposable elements in cancer include the inactivation of tumor suppressor genes, the overexpression of oncogenes, and the synthesis of chimeric proteins with oncogenic features [7]. Conversely, the re-expression of tumor-suppressing genes, the stimulation of anti-viral and immune responses, and the establishment of tumor-specific neoantigens are among the anti-tumoral activities of ERVs and LTRs [7], and constitute a targetable vulnerability.

Intriguingly, TEs are often overexpressed in tumors compared with healthy tissue, and this observation has prompted the search for anti-TE T-cell responses in cancer [7]. Upon insertions in the genome, LTRs are frequently marked by repressive epigenetic modifications to suppress their expression. These modifications induce local chromatin modification that can spread to the surrounding DNA and modulate the expression of adjacent genes in a tissue-specific manner. Beyond the epigenetic modifications, a large proportion of LTRs contain binding motifs for sequence-specific TFs, which can promote or enhance the expression of specific genes [8]. They are very interesting elements with different functions because they can produce both functional epitopes and activate specific molecular pathways [9]. LTRs contain important gene regulatory sequence elements, such as promoters and enhancers, and when inserted in proximity to a host gene, they can influence its expression pattern [10]. In addition to their transcriptional activity, the insertion of these elements into the human genome has allowed the formation of new transcription start sites (TSSs) in several genes. Among LTRs, the LTR12 family is found more frequently near gene promoters and TSSs than the others, and evidence suggests that LTR12C has the highest enrichment value [11]. The LTR12 family is classified as a part of the ERV-9 family, and it includes elements like LTR12C, LTR12D, and LTR12E. So far, it has been observed that LTR12C induces the expression of several genes, including TP63; specifically, it induces the tissue-specific expression of GTA-p63 (a testicular isoform of the p53 tumor suppressor homologue p63 derived from the gene TP63). In male germ cells, GTA-p63 is capable of inducing apoptosis upon genotoxic stress to protect the germline from genome damage. This activation produces a slightly different protein called GTA-p63 that contains part of the LTR sequence in the first exon of its precursor mRNA. LTR insertion in the human genome allows the formation of a new p63 protein isoform in healthy human testis, which is strongly expressed in spermatogenic precursors but not in mature spermatozoa [12]. In response to DNA damage, GTA-p63 is activated by caspase cleavage near its carboxyterminal domain, thus suppressing proliferation and inducing apoptosis in male germ cells [12]. The insertion of the ERV-LTR upstream of TP63 represents an interesting example of an ERV-derived sequence that controls the expression of a gene that functions in germ-line protection. Testicular cancer cells are capable of silencing ERV-LTR promoter activity, suppressing its potential tumor-suppressive mechanism [12]. However, the pharmacological inhibition of DNA methyltransferase (DNMT) and histone deacetylase (HDAC) completely restores the GTA-p63 expression in testicular cancer cells and, in combination with standard cancer treatment, cooperate to enhance cancer cell death [12]. We reasoned that the activation of GTA-p63 could be restored to also induce the apoptotic pathway in GBM and, to explore this possibility, we activated the LTR transposable elements in two GBM cell lines, T98-G and U87-MG. We also examined other genes whose expressions were modulated by TP63 or LTR12C activation and the viability of the cancer cell lines after treatment.

## 2. Materials and Methods

### 2.1. Cell Culture and Treatments

Human glioma U87-MG (ATCC^®^ HTB14™, American Type Culture Collection, Manassas, VA, USA) and glioblastoma T98-G (ATCC^®^ CRL1690™, American Type Culture Collection, Manassas, VA, USA) cells were cultured in Eagle’s Minimum Essential Medium (E-MEM, Sigma-Aldrich D-4655, St. Louis, MO, USA); supplemented with 10% FBS (Gibco, A5256701, Thermo Fisher Scientific, Waltham, MA, USA), 10.000 U/mL penicillin (Aurogene), 10.000 U/mL streptomycin (Aurogene), and 10 mM non-essential amino acids (Gibco, 11140050, Thermo Fisher Scientific, Waltham, MA, USA); and maintained at 37 °C in 5% CO_2_ in an air-humified incubator. The hypomethylating agent 5-aza-2′-deoxycitidine (DAC, Selleckchem, S1200, Houston, TX, USA) was resuspended to a final concentration of 100 mM in Dimethyl sulfoxide (DMSO); the pan-HDAC inhibitor SB939 (Selleckchem, S1515, Houston, TX, USA) was resuspended in DMSO to reach a final concentration of 50 mM and subsequently used to treat the cells.

Exponentially growing cells were treated with 500 nM DAC for 96 h, 500 nM SB939 for 18 h, or a combination of both (500 nM DAC for 78 h, then 500 nM of both drugs for the remaining 18 h). The different condition was chosen for two key reasons: first to maintain consistency with the experimental conditions previously set by Goyal et al. [5], and second, due to the fact that DNMT1 inhibitors generally need to be incorporated into DNA to have an effect, they needed more time to exert their activity compared with the HDAC inhibitors.

### 2.2. IncuCyte^®^ Live-Cell Analysis

Incucyte^®^ Nuclight Rapid NIR Dye for Live-Cell Nuclear Labeling (Sartorius, 4804) and Incucyte^®^ Cytotox Dye for Counting Dead Cells (Sartorius, 4633, Göttingen, Germany) were added to the culture medium according to the manufacturer’s instructions. The cells were monitored daily, and images were analyzed using the IncuCyte^®^ Live-Cell Analysis System (Sartorius, Göttingen, Germany) to calculate the number of living and dead cells in each well, which were normalized to the number of cells at the initial time point.

### 2.3. Bioinformatics Analysis

Publicly available RNA-sequence data of T98-G cells treated with 500 nM DAC and 500 nM SB939 or DMSO (GSE209772, [5]) was analyzed using the Geo2R web tool, setting the thresholds for differential gene expressions for the Log_2_(fold change) and False Discovery Rate (FDR) to 1 and 0.05, respectively. A Gene Ontology analysis of differentially expressed genes was performed in the R environment using the topGO package (version 2.61.0, Bioconductor, Seattle, WA, USA) [13], with the list of all expressed genes in T98-G cells used as a background for the analysis. The statistical significance of GO terms was calculated using the weightFisher function of the package. For the Gene Set Enrichment Analysis (GSEA), all expressed genes were ranked according to log2(fold change) and analyzed against the cancer hallmark gene sets from MSigDB using clusterProfiler [14] in the R environment by calculating the normalized enrichment score (NES) for each gene set based on 1.000 permutations. The analysis of the splice site strength was performed using the MaxEntScan web tool [15]. An in silico translation of different TP63 mRNA variants was performed using the Translate webtool from the SIB Swiss Institute of Bioinformatics [16].

### 2.4. Total RNA Extraction

Total RNA was extracted from cells using TRIzol reagent (Invitrogen, 15596018, Carlsbad, CA, USA) following the manufacturer’s instructions. Genomic DNA was enzymatically digested by incubating the samples with RNase-free DNase (Ambion, AM2222, hermo Fisher Scientific, Waltham, MA, USA) for 1 h at 37 °C. Subsequently, RNA was isolated by phenol-chloroform extraction (Merck, P1944, Darmstadt, Germany).

### 2.5. RT-PCR and RT-qPCR Analyses

An amount of 1 μg of total RNA was reverse-transcribed using random primers (Merck, 11034731001, Darmstadt, Germany) and M-MLV reverse transcriptase (Promega, M1701, Madison, WI, USA) according to the manufacturer’s instructions. A total of 15 ng of cDNA in a final volume of 20 μL was amplified by conventional polymerase chain reaction (PCR) using GoTaq polymerase (Promega, M7841, Madison, WI, USA) and analyzed by agarose gel electrophoresis. Amplified DNA was isolated from the agarose gel using a QIAquick Gel Extraction Kit (QIAGEN, Hilden, Germany) and processed for Sanger sequencing with the Mix2Seq Kit (Eurofins, Nantes, Brussels) according to the manufacturer’s instructions. Quantitative PCR (qPCR) analysis was performed on a QuantStudio3 Real-Time PCR System (Applied Biosystems, Waltham, MA, USA) using 25 ng of cDNA in a final volume of 20 μL with PowerUp™ SYBR™ Green Master Mix (ThermoFisher, A25742, Waltham, MA, USA), following the manufacturer’s protocol. Differences in gene expression were calculated using the 2^−ΔΔCt^ method and RPL34 as the housekeeping gene. The primers used in this study are listed in Appendix A [5,12,17].

### 2.6. SDS-PAGE and Western Blot

Cell lysates were prepared in RIPA buffer (50 mM Tris pH 7.4, 1% NP-40, 0.5% sodium deoxycholate, 0.1% SDS, 150 mM NaCl, 1 mM EDTA, 0.5 mM Na_3_VO_4_, and 1 mM DTT) supplemented with a protease inhibitor cocktail (Merck, 539132, Darmstadt, Germany). Following sonication, the samples were centrifuged at 13.000 rpm for 10 min at 4 °C. In total, 10 μg of protein was resolved by SDS-PAGE and transferred onto a PVDF membrane (Amersham, Merck GE10600023, Darmstadt, Germany). After incubating 5% non-fat dry milk in a PBS blocking solution for 1 h at 25 °C, membranes were incubated overnight at 4 °C with primary antibodies and used in a 1:1.000 dilution in 5% BSA in PBS. The primary antibodies used for this study were the following: mouse anti-HSP90 (Santa Cruz, sc-101494, Dallas, TX, USA), mouse anti-GAPDH (Santa Cruz, sc-47724), mouse anti-SOD1 (Santa Cruz, sc-17767), mouse anti-p53 (DO-1, Santa Cruz, sc-126), rabbit anti-p63 (D2K8X, Cell signaling, 13109S), rabbit anti-Pan TP63 (D9L7L, Cell Signaling, 39692), rabbit anti-CDKN1A (Cell Signaling, 2947, Danvers, MA, USA), rabbit anti CDK6 (Cell Signaling, D4S8S), and rabbit anti-PARP1 (Cell Signaling, 9542S). The next day, after 4 5 min washes with 0.5% Tween-PBS, the blots were incubated with secondary antibodies (anti-rabbit and anti-mouse HRP-linked secondary antibodies, Jackson ImmunoResearch, West Grove, PA, USA) at a 1:10.000 dilution in 5% non-fat dry milk in PBS. The signal was developed using the Clarity Western ECL Blotting Substrate (Biorad, 1705061, Hercules, CA, USA), and images were acquired with a UVITEC system. (Uvitec, Cambridge, UK)

### 2.7. Quantification and Statistical Analysis

All data are expressed as the mean ± SD. The unpaired Student’s *t*-test and one-way ANOVA were performed using Graphpad Prism software (version 9.5.1, GraphPad Software, San Diego, CA, USA). When needed, the *p* values were adjusted for multiple testing with the Bonferroni method. A *p* value < 0.05 was considered statistically significant. The half-maximal inhibitory concentration (IC50) values of DAC, SB939, and DAC + SB939 after 96 h treatments were calculated based on Incucyte Live-Cell Analysis data using non-linear regression analysis (absolute IC_50_ calculation) in Graphpad Prism. Densitometric analyses of the Western blot images were performed using ImageJ software (version 1.3K, NIH, Bethesda, MD, USA). IGV software (version 2.15.4; Broad Institute, Cambridge, MA, USA) [18] was used to visualize the gene body coverage of RNA-seq reads and create read coverage images for the TP63 gene. Enrichment plots were created using ggplot2 [19] and enrichplot [20] R package version 1.12.1. Volcano plots were generated using the EnhancedVolcano package in the R environment [21]. Other plots were created with Graphpad Prism.

## 3. Results

### 3.1. DNA Demethylation and HDAC Inhibition Reactivated LTR12C Retroviral Element Expression in GBM Cell Lines

Extensive evidence reports aberrant epigenetic regulation in GBM, leading to the reactivation of different families of retroviral elements [22]. The reactivation of these elements, including the LTR12C subfamily, can be exacerbated after treatment with epigenetic drugs, such as DNA methyltransferase (DNMT) and HDACs inhibitors [23]. Based on this knowledge, we inhibited both DNMT1 and HDACs in the GBM cell lines U87-MG and T98-G to evaluate the effect on the expression levels of LTR12C and of genes regulated by this retroviral element. The DNMT1 inhibitor DAC and the HDAC inhibitor Pracinostat (SB939) were used either individually or in combination as specified in the Materials and Methods Section (Appendix A). The analyses by RT-qPCR indicated that the combined inhibition of DNMT1 and HDACs strongly induced the expression of LTR12C, with 20-fold and 84-fold increases in U87-MG (Appendix A) and T98-G (Appendix A) cells, respectively. Single treatments with DAC or SB939 produced a much milder effect (Appendix A), suggesting a synergistic mechanism behind the reactivation of LTR12C after the sequential inhibition of DNMT1 and HDACs.

To further explore the transcriptional programs modulated by DNA global demethylation and HDACs inhibition in GBM, we analyzed previously published RNA-seq data (GSE209772) derived from T98-G cells treated with 500 nM DAC and 500 nM SB939 [5], focusing on differential gene expression. Considering a minimum absolute fold change (abs FC) of 2 and a False Discovery Rate (*p* adj) threshold of 0.05, we detected a total of 7228 differentially expressed genes (DEGs), representing 37.86% of all genes expressed in this cell line. Among these genes, 29.01% were downregulated, while 70.19% were upregulated (Figure 1A,B). The most differentially expressed and significant genes were predominantly upregulated, as shown in the volcano plot in Figure 1C. The Gene Set Enrichment Analysis (GSEA), which determines whether a predefined set of genes shows concordant and statistically significant differences between two biological states, indicated the activation of nine gene sets including spermatogenesis, KRAS signaling downregulation, and estrogen response (early and late) among the most significant (Figure 1D–F).

The Gene Ontology analysis highlighted that the ten most enriched biological process terms were related to processes such as cell adhesion, nervous system processes, the G protein-coupled receptor signaling pathway and transmembrane transport, and piRNA processing (Figure 1G). The molecular function and cellular component categories indicated that many genes were involved in signal receptor activity and the extracellular region, respectively (Appendix A). The downregulated genes were associated with mitochondrial translation and respiratory chain complex assembly (Figure 1H).

To validate these data, we performed an independent RT-qPCR analysis of two genes associated with the cell adhesion category in the biological process ontology: TIAM Rac1-associated GEF 1 (TIAM1) and Dual Specificity Phosphatase 1 (DUSP1). The results confirmed their upregulation after treatment with DAC and SB939, both in the T98-G and U87-MG cell lines, indicating that these observations are reproducible in different GBM cell models. Specifically, the TIAM1 expression increased more than fourfold in the T98-G cell line (Figure 1I), while the DUSP1 expression increased 20-fold (Figure 1J). In the U87-MG cells, TIAM1 expression doubled (Figure 1K), while the mRNA levels of DUSP1 increased nearly threefold (Figure 1L). Furthermore, the combined treatment with DAC and SB939 produced stronger effects on the DUSP1 and TIAM1 mRNA expressions compared with DAC or SB939 alone, in both the T98-G (Figure 1I,J) and U87-MG (Figure 1K,L), suggesting an additive effect of these two drugs.

Taken together, these data show that the most significantly altered genes after the DNMT1 and HDAC inhibitions in these two cell lines were linked to biological processes that occurred on the membrane surface and consequently on the extracellular aspect involved in cell adhesion.

### 3.2. DNA Demethylation and HDAC Inhibition Induced the Expression of LTR12C-Driven Genes in Glioblastoma Cell Lines

As previously explained, the combined inhibition of DNMT1 and HDACs reactivate the molecular mechanisms underlying the expression of retroviral elements, including the LTR12C subfamily [5,11]. For this reason, we wondered whether the expression of any LTR12C-driven protein-coding gene could be altered by these epi-drugs in T98-G cells.

To answer this question, we analyzed the pan-cancer de novo assembly transcriptome, which reported deregulated transcripts after the dual inhibition of DNMT1 and HDACs in cancer cell lines and the distance of the nearest repetitive element from the TSS [5]. By matching all the annotated transcripts containing a LTR12C under 5.000 bp from the TSS and the deregulated genes after combined treatment with DNMT1 and HDAC inhibitors, we obtained 300 genes (Appendix A). Among these, 235 genes were upregulated (Figure 2A) and 65 were downregulated (Figure 2B). The functional annotation of this list indicated that 45 transcripts belonged to the category of non-coding genes while the remaining 255 were annotated as coding genes (Figure 2C). We further analyzed the 255 protein-coding genes included in the list (Figure 2C) by manually checking their sequences and found that only 12 of them had an LTR12C that overlapped their TSS (or at least the TSS of one of their transcriptional variants) (Figure 2D). In this regard, Othani et al. identified a list of LTR12C-driven genes in different cancer cell lines characterized by a higher expression of read-through RNA transcripts due to the presence of H4K4me3 and H3K27ac markers [24]. Comparing our 12 genes with their gene list, we found that KCNN2, PPP1R3A, and ACSBG1 belonged to this category.

To assess any possible shared function of our 12 genes, we examined the NCBI gene summaries of these genes and determined that they were not involved in a single specific cellular process, as they may have different functions. For example, GBP5, GBP2, and Immunity Related GTPase M (IRGM) are involved in the immune response, Pyroglutamyl-Peptidase I Like (PGPEP1L) and Ribosomal Protein L3 Like (RPL3L) are involved in protein metabolism, and SEMA4D and Acyl-CoA Synthetase Bubblegum Family Member 1 (ACSBG1) are involved in neuronal development and myelinogenesis. In contrast, TP63 is involved in the regulation of apoptosis during skin and germ cell differentiation, and some TP63 isoforms can protect the germ line by eliminating oocytes or testicular germ cells that have suffered DNA damage [12,25].

We then wondered whether any of these genes could play a role in the biology of GBM. Previous studies reported that GBP2 and GBP5 have a pro-tumoral role [26,27], while some TP63 isoforms are involved in the DNA damage response and have already been studied in GBM [28,29]. Interestingly, TP63 expression can be induced by temozolomide, inhibiting cell proliferation [28]. So, we validated the expression changes of these genes in our independent set of experiments in the T98-G and U87-MG cell lines. The analysis by RT-qPCR indicated that the GBP2 expression was slightly upregulated in the T98-G cells (Figure 2E), while we observed an increasing trend in the U87-MG cells (Figure 2F). Instead, GBP5 and TP63 were strongly upregulated in both cell lines. Regarding TP63, both in the T98-G (Figure 2E) and U87-MG (Figure 2F) cell lines, the inhibition of DNMT1 alone was sufficient to increase its expressions by 54 and 35 times, respectively, even though statistical significance was not reached. The combined inhibition of DNMT1 and HDAC further induced the TP63 expression, which led to a 229-fold increase in the T98-G cells (Figure 2E) and a 175-fold increase in U87-MG cells (Figure 2F). Overall, this data indicates a synergistic effect of the two drugs on the regulation of these genes. Given its biological role in stemness maintenance, differentiation, and the activation of apoptotic mechanisms after genotoxic insults, we decided to focus our attention on TP63 for further investigation.

#### 3.2.1. LTR-Driven GTA-p63 Expression in Glioblastoma Cell Models

TP63 is rarely mutated in cancer but is often overexpressed and/or amplified in specific cancerous settings [30]. In fact, there are several transcription variants of TP63, each with specific functions, and the usage of one of three alternative promoters can lead to proteins with different N-terminal extremities. Among these, TA-p63 and the LTR-driven GTA-p63 exert pro-apoptotic functions, while the truncated variant ∆N-p63 performs pro-survival roles [12,31]. As mentioned earlier, the transcription start site (TSS) for GTA-p63 is embedded within a LTR12C element [12].

Alternative splicing events can also produce protein variants that differ in their C-terminal portions, resulting in transcripts of different lengths. In fact, the α isoform constitutes the full-length protein, while β, γ, δ, and ε correspond to different C-terminal truncated proteins [31] (Figure 3A).

According to the RNA-seq data discussed above, combinatory treatment with DAC and SB939 induced a 146-fold increase in the TP63 mRNA levels in the T98-G cells. So, we hypothesized that the dual inhibition of DNMT1 and HDAC could induce TP63 expression by reactivating the specific LTR12C on the TSS of the GTA-p63 variant. Indeed, looking at the body coverage of the RNA-seq reads, we inferred that this treatment reactivated the LTR12C alternative promoter but not the promoter of the TA-p63 variant (Figure 3B). This would be consistent with the ability of these drugs to induce LTR12C elements, resulting in the augmented expression of their downstream genes.

To assess whether the reactivation of the alternative isoform guided by LTR12C could also lead to an increase in the p63 protein, we performed a Western blot analysis of the T98-G cells treated with DAC and/or SB939. Unfortunately, although TA-p63α and GTA-p63 differ by 19 amino acids at the N-terminal end, there is still no antibody that recognizes only the GTA variant. For this reason, we used two different antibodies for our analyses: the first targets all the p63 *α* isoforms (TA-p63*α*, GTA-p63, and ∆N-p63*α*) (Figure 3E, upper panel), while the second one recognizes all the possible isoforms of p63, including those with the C-terminal truncated (PAN p63) (Figure 3E, lower panel). The results indicate that the inhibition of DNMT1 alone slightly induced p63*α* expression, and densitometric analyses showed that the sequential inhibition of DNMT1 and HDAC increased the p63 protein level by 124 times compared with control cells (Figure 3F).

To explore whether the increased expression of the total TP63 mRNA levels after treatment was actually due to specific induction of the GTA-p63 isoform, we performed RT-qPCR analyses, using different primer pairs (Figure 3A) to amplify them: the LTR12C-driven isoform GTA-p63, the TA-p63*α* mRNA variant, and all TP63*α* mRNA variants to assess the TP63*α* global expression changes.

The results show that GTA-p63 was upregulated after the combinatory treatment with DAC and SB939 in both cell lines, with up to a thousand-fold increase in comparison with the control (FC > 5000 in the U87-MG and FC > 1270 in the T98-G cells) (Figure 3C,D). In the TA-p63*α* isoform, the co-treatment caused a much smaller increase in the expression than in the GTA-p63, which resulted in a maximum of 40.17 times the controls in the U87-MG cell line (Figure 3C,D).

Furthermore, we confirmed that the combined inhibition of DNMT1 and HDACs induced *α* isoforms of TP63 by amplifying the two terminal exons (13 and 14) of these mRNA variants. In this case, the cotreatment with DAC and SB939 led to an induction of global TP63*α* expression, which reached an increase of 226.5 times in the U87-MG cells (Figure 3C) and an increase of 275 times in the T98-G cells (Figure 3D). Collectively, these data indicate that DNA demethylation alone increased the global TP63*α* expression, and the combination of DNMT1 and HDAC inhibitors exacerbated this effect in both the U87-MG and T98-G cells. Interestingly, the RT-qPCR analyses demonstrated that these molecules induced the pro-apoptotic isoform of GTA-p63, which was expressed hundreds of times higher than the canonical form. All together, these data suggest a synergistic effect between the DNMT1 and HDAC inhibitors on the induction of the LTR12C-driven TP63*α* transcript variant, GTA-p63, which accounted for the greatest part of the global expression changes in the TP63 mRNA levels.

#### 3.2.2. Characterization of GTA-p63 Transcripts in T98-G Cell Line

Three unique exons are located upstream of the canonical exon 1 of the human TP63 gene (U1, U2, and U3) (Figure 3A). Splicing one or more of these exons directly to the previously described exon 2 leads to the skipping of exon 1. The unique p63 isoforms are readily detectable in human testicular tissues, and the direct fusion of exon U1 with exon 2 constitutes by far the predominant splice product [12]. The main predicted GTA-p63 protein sequence (exon U1 to exon 2) differs from the canonical TA-p63 protein isoform (named TA) by only the first 19 aminoacidic residues [12].

Specifically, the TSS of the GTA-p63 gene matches the sequence corresponding to the endogenous retrovirus 9 (ERV9) LTR, starting from the R-U5 region located immediately next to the 3′ of the LTR promoter (Figure 3A). In the literature it has been described that a minority of TP63 gene transcripts may be spliced in exons U2 or U3 (which are mutually exclusive) between exons U1 and 2 [12].

To characterize all the possible transcripts produced in the T98-G cells after the treatment with DAC and SB939, we amplified the first exons of the gene by using primers located in exon U1 and exon 2 (Figure 3A). The results show the presence of two products, which differed by about 100 base pairs (Appendix A); Sanger sequencing indicated that they corresponded to the U1-EX2 and U1-U2-EX2 isoforms. Since the two products appeared to be equivalent in quantity, we calculated the scores of the 3′ junction sites in U2 and exon 2 by means of the MatEntScan tool [32]; the resulting values are very similar to each other (3′ss U2 = 9.42; 3′ss EX2 = 9.53), suggesting that the bond of the two exons with U1 could have the same strength. At this point, we decided to define the possible initiation of the translation process in the two different GTA-p63 isoforms (including exon U2 or not). By observing the TP63 gene in the Genome Browser [33], we noticed that the translation of the GTA-p63 isoform started from the first ATG codon, that is, in exon U1, between the LTR12C promoter and a LINE element at the 3′ (Appendix A).

The donor splice site of U1, which permits the match with exon 2 or exon U2, was included in the LINE element. We calculated that the score of this 5′ splice site corresponded to 10.47; interestingly this value was higher than that of the canonical exon 1 donor site, which corresponded to 8.56. As explained above, the first 19 amino acids of the GTA-p63 protein were different from those present in the conventional TA-p63 protein. In detail, 16 of the 19 amino acids derived from exon U1 were encoded by the LINE DNA, while the last C (Cysteine) was encoded both by LINE and exon 2/U2 (Appendix A). Having observed that the transcript that included the exons U1 and U2 was present in the same quantity as the transcript U1-EX2, we tried to translate the protein in silico to check whether it was in frame with the rest of TA-p63 protein. Using the Expasy translate tool, we obtained two possible open reading frames (ORFs): The first started from the same ATG of the U1-EX2 isoform but had an early stop codon that allowed the production of a short peptide containing 18 aa. In the second ORF, the ATG starting codon, which allowed this translation, was included in the exon U2 that encodes the first 20 amino acids; interestingly, from the 21st amino acid onwards, the protein was equal to TA-p63 (Appendix A).

### 3.3. The Expression of Apoptosis-Related Genes in Glioblastoma Cell Models Could Be Affected by DNMT1 and HDAC Inhibition

DNA methyltransferase (DNMT) and histone deacetylase (HDAC) inhibitors can constrain cell growth and induce apoptosis. In fact, HDACs play a major role in carcinogenesis through several pathways, and HDAC inhibitors can inhibit HDAC activity by various mechanisms, resulting in cell cycle arrest, cell growth inhibition, and apoptosis induction [34]. Among the pro-apoptotic genes, CDKN1A (p21) can be induced after HDAC inhibition [35]. P21 is involved in the regulation of the cell cycle and senescence and is a CDK inhibitor that physically interacts with CDK4 and CDK6, blocking the cell cycle in G1 phase and inducing quiescence and cell senescence [36]; furthermore, the p21 gene is transcriptionally regulated by TP63 [37].

For this reason, we decided to investigate how the DNMT1 and HDAC combined inhibition could act on this pathway and whether the two drugs produced different effects than the individual treatments. To do this, we evaluated the expression levels of p21, CDK4, and CDK6 by qRT-PCR. The results show that the co-treatment with DAC and SB939 induced p21 expression at the mRNA level in both the U87-MG and T98-G cell lines (Figure 4A), with 4- and 12-fold increases, respectively. In the U87-MG cells, the treatment with DAC increased the p21 expression by 3 times compared with the control cells (Figure 4A), whereas in the T98-G cells, only the double treatment was able to increase the p21 expression by 12 times (Figure 4A). The CDK4 gene expression was downregulated after the double treatment by 0.44 times in the U87-MG and 0.62 times in the T98-G cells (Figure 4B,C), while CDK6 was not affected at all after the combinatory treatment with epi-drugs (Figure 4B,C). At the protein level, we observed a significant upregulation of p21 in the T98-G cells treated with SB939 alone, and a trend towards increased expression in samples treated with the combination of the two drugs (Figure 4D). However, this effect was not recapitulated in the U87-MG cell line (Figure 4D). Taken together, these results show that the inhibition of DNMT1 and HDACs affected the expression of downstream effectors of TP63, such as p21, at least at the mRNA level, which possibly contributed to the cell cycle regulation.

To better define the pro-apoptotic genes activation in our system, we considered that both TA-p63α and GTA-p63 can induce the transcription of classical p53 target genes. In fact, in addition to p21, they can also induce BCL2 Binding Component 3 (BBC3, or PUMA) and Phorbol-12-Myristate-13-Acetate-Induced Protein 1 (PMAIP, or NOXA) [12,31]. For this reason, we assessed the mRNA levels of PUMA and NOXA. In greater detail, the PUMA mRNA levels doubled after the treatment with DAC, while the combination with SB939 caused an increase of 3 times compared with the control cells; instead, the NOXA expression levels were augmented by almost 3 times after the inhibition of both DNMT1 and HDACs (Figure 5A). In the T98-G cells, the PUMA expression did not change after the double DAC and SB939 treatment, but we observed a higher expression level of NOXA mRNA (FC = 3) than in the samples treated only with DAC or SB939 (Figure 5B). Finally, we analyzed a late apoptosis marker, the cleavage of PARP1, by Western blot analysis and observed an increase in this cleavage after the combined inhibition of DNMT1 and HDACs in both cell lines (Figure 5C). Furthermore, in the U87-MG cells, this effect was stronger than observed in samples treated only with DAC or SB939 (Figure 5C). Thus, the combined inhibition of DNMT1 and HDAC was able to slightly induce apoptosis in these glioblastoma cell models. Since the T98-G cell line carries an inactivating mutation of TP53 [38], the upregulations of p21, PUMA, and NOXA following the inhibition of DNMT1 and HDACs could be mediated by p63 and not p53. Moreover, it has been reported that mutant p53 can influence p63 activity, both inhibiting the transcriptional regulation mediated by the anti-tumoral TA isoforms and inducing the expression of the pro-tumoral ∆N-p63 [39]. For this reason, we analyzed mutant p53 protein levels in our samples of T98-G cells by Western blot. The results indicate a reduction in p53 protein levels in the T98-G cells after treatment with SB939 (with a fold reduction of almost 20% compared with the vehicle) and after cotreatment with DAC and SB939 (fold reduction of 25% compared with the vehicle) (Figure 5D). Taken together, these data suggest that the sequential inhibition of DNMT1 and HDACs leads to the upregulation of markers of cell cycle arrest and apoptosis and that this event may be p53-independent, at least in the T98-G cell line.

### 3.4. DNA Demethylation and HDAC Inhibition Reduced the Viability of the Glioblastoma Cell Lines

Once it was found that the DNMT1-inhibitor, DAC, and the pan-HDAC inhibitor, SB939, could induce p21 expression and PARP1 cleavage in the U87-MG and T98-G cells, we decided to verify the effect of these drugs on the cell viability by MTS assay and Incucyte Live-Cell Analysis.

The results of the MTS analysis indicated that in the U87-MG cell line, the treatment with 500 nM DAC did not cause a significant reduction in the cell viability, while 500 nM SB939 reduced the viability by 25% compared with the control (Figure 6A). Furthermore, the combination of DAC and SB939 led to a significant reduction of 35% compared with the control, which slightly exacerbated the effect of the two drugs used alone (Figure 6A). While the T98-G cell line did not respond to the single treatments, the combination of DNMT1 and HDACs inhibitors resulted in a significant reduction in the cell viability of 40% compared with the control group (Figure 6A), suggesting a synergistic effect of the two inhibitors in this cell line.

To better clarify the phenotypic effect of these drugs in glioblastoma cell lines, we performed a cell viability assay by means of Incucyte^®^ Live-Cell Analysis using specific fluorescent reagents to count the number of living and dying cells. We performed a dose of DAC, SB939, and their combination, with concentrations between 100 nM and 2.500 nM. After 96 h, SB939 was less effective than DAC at reducing the cell viability in both the cell lines (Figure 6B). The combination of the two drugs exerted synergic effects on the cell proliferation in both the cell lines. The half-maximal inhibitory concentrations (IC50) were 406 nM in the U87-MG cells and 175 nM in the T98-G cells for DAC + SB939. The IC50 values of the single treatments and combination index of the two drugs are reported in Table 1.

Next, we performed the Incucyte Live-Cell assay in which the U87-MG and T98-G cells were monitored daily for four days after the treatment. The results indicate that the treatment with 500 nM SB939 did not significantly affect the total number of cells per well in both the cell lines. Instead, the treatment with 500 nM DAC reduced the cell growth by half or more compared with the control, starting from 48 h after the treatment (Figure 6C). Furthermore, the effect of the combinatory inhibition of DNMT1 and HDACs on the cell growth was comparable with the DNMT1 inhibition alone (Figure 6C), suggesting the lack of an additive effect of the two molecules on the cell proliferation for both cell lines. Then, we evaluated the number of dying cells by adding the Cytotoxic Green Reagent (Sartorius) to the media during the experimental time. In this way, we detected an increasing trend in the number of dying cells for the U87-MG cells treated with DAC and SB939 (Figure 6D), and for the T98-G cells treated only with DAC or with the combination of the two molecules (Figure 6D). Taken together, this data suggests that DNMT1 and HDACs sequential inhibition can affect the cell growth and viability in different glioblastoma cell models, and that DAC, but not SB939, contributed the most to this effect.

## 4. Discussion

Glioblastoma remains an extremely aggressive and deadly tumor. Despite advances in surgery and adjuvant therapies, the high heterogeneity, aggressiveness, and tendency to recur severely limit therapeutic choices for this disease.

Non-coding DNAs, especially transposable elements (TEs), have emerged as new participants in cancer biology. During the last few years, different studies have demonstrated that TEs harbor cis-regulatory sequences for human transcription factors and can thus contribute to the human gene regulatory elements, such as promoters and enhancers [40].

While in normal conditions, TEs are silenced, it is becoming increasingly clear that they can be epigenetically activated in different conditions, including brain tumors [41].

Among the TEs, solo long terminal repeats (solo-LTRs) were exapted as *cis*-regulatory regions (e.g., promoters, transcription factors binding sites) of human genes [42]. For this reason, their activity is strictly regulated by epigenetic modifications to prevent aberrant gene expression [43]. This delicate balance is lost in cancer, where global DNA hypomethylation leads to an increased expression of endogenous retroviruses (ERVs) and their deriving LTRs [44]. From this perspective, the LTR12 subfamily is particularly interesting for two reasons. The first is that LTR12 elements overlap with TSS promoter genes more frequently than other LTRs [11]; therefore, epi-drugs may affect the expression of genes regulated by the LTR12 subfamily of transposable elements. The second is that the reactivation of LTR12C elements leads to the production of tumor-specific neoantigens, thus offering new targets for immunotherapy [5].

Notably, the combined inhibition of DNMT1 and HDACs can induce a stronger reactivation of the LTR12C subfamily of retroviral elements compared with the inhibition of only one of these protein families [11]. Since similar observations have also been made in several GBM cell models [5], we aimed to study the effect of epi-drugs on the regulation of protein-coding genes, focusing on the TP63 gene, whose expression can start from LTR12C promoters in glioblastoma cell lines.

To understand which pathways were activated following the treatments, we performed a Gene Set Enrichment Analysis (GSEA) on the differentially expressed genes (DEGs) after the epigenetic treatment with a combination of DNMT1 and HDAC-inhibitors [5]. Among the resulting gene sets, the presence of two categories related to KRAS signaling was interesting since this signaling plays a crucial role in glioblastoma, even if mutations in the KRAS gene are rare in human gliomas.

In particular, KRAS is involved in the regulation of sensitivity to platinum-based chemotherapy, such as cisplatin, via the RAF/MEK/ERK signaling pathway. The endogenous expression of KRAS is increased in response to cisplatin treatment, and the inhibition of MEK may modulate the sensitivity of glioblastoma cells to cisplatin. In addition, the ectopic expression of a constitutively active form of KRAS (KRASG12V) may influence the apoptotic response of glioblastoma cells to cisplatin treatment [45]. Interestingly, mutant KRAS regulates interferon-stimulated gene expression and transposable elements, such as LINE and LTR, including the LTR12C subfamily in transformed lung cells [46], possibly linking the cancer response to chemotherapy to the activation of transposable elements and the inflammation pathway.

The increased expression of genes related to the development of male germ cells (Hallmark Spermatogenesis) suggests that the combination of DAC and SB939 may increase the expression of Cancer–Testis Antigens (CTAs) in glioblastoma cell lines. The enrichment of genes associated with spermatogenesis is due to the disruption of silencing mechanisms, such as methylation and deacetylation, rather than simply background noise occurring after treatment.The expression of CTAs is desirable to find tumor-specific neoantigens to be utilized for the development of vaccines. In this regard, studies have demonstrated that DNMT and HDAC inhibitors operate synergistically on the induction of CTAs [47], suggesting that a similar mechanism might also be reproducible in glioblastoma. Furthermore, the upregulation of genes related to immune response after the double inhibition of DNMT1 and HDACs is consistent with the evidence that DNA hypomethylation triggers viral mimicry against ERV-derived dsRNAs [48].

Since our analysis focused on the activation of LTR12C elements, we analyzed the GEO dataset GSE209772 to find possible LTR-driven genes among the differentially expressed genes (DEGs). As we detailed in the Results Section, we selected 12 genes that contained an LTR12C in or near their transcription start site (TSS).

According to the literature, among the genes whose expressions change after treatment and have an LTR12C element at the TSS site, only IRGM, GBP2, GBP5, and TP63 are known to play a meaningful role in glioblastoma. GBP2 and GBP5 are interferon-inducible guanylate-binding proteins, acting in innate immunity and exerting a tumor-sustaining function in glioblastoma. Indeed, both genes may be overexpressed in glioblastoma compared with healthy tissue, and higher levels of GBP5 expression are correlated with a worse prognosis [26,27]. IRGM can influence glioma progression through its involvement in the regulation of both autophagy and inflammation and, while the exact relationship between IRGM and glioma is still being investigated, it appears that IRGM’s functions in immune modulation, autophagy, and inflammation may play important roles in glioma tumor biology [49,50].

Among these genes, TP63 is of particular significance due to its isoform-specific effects and role in LTR-induced transcription.

TP63, a member of the TP53 family of transcription factors, plays opposite roles in cancer depending on which of its isoforms is expressed within cells: GTA-p63 and TA-p63 have pro-apoptotic and potential anti-tumoral effects, while ∆N-p63 promotes stemness and plays a pro-tumoral function [51,52].

In glioblastoma there is little data about the effects of the p63 protein, but the increased expression of TA-p63 inhibits the expression of a well-known oncogene, MYC, reducing the cell proliferation and invasiveness [28]. Only recently, the male germinal isoform of p63, GTA-p63, has been considered in the glioblastoma field; indeed, antigenic TE-derived peptides have been observed after a combination treatment with decitabine and panobinostat in glioblastoma cell lines [53].

Subsequently, we investigated whether the activation of LTR12C was specifically responsible for the induction of GTA-p63. Our data show that DAC and SB939 could act synergistically to induce GTA-p63 expression in both the U87-MG and T98-G cell lines, which led to greater overexpression compared with samples treated only with DAC or SB939. This is in line with the idea that these epigenetic-modifying drugs act on different molecular pathways to reach the same end target and eventually amplify each other’s effects. Notably, we also observed that combined treatment with epi-drugs, having a wide spectrum of endpoints, also led to the overexpression of oncogenes, such as GBP2 and GBP5. These observations are consistent with the notion that LTRs, when overexpressed after DNA hypomethylation and HDAC inhibition, may play a dual role in cancer, with both pro- and anti-tumoral effects.

The induction of TP63 expression is particularly representative of the effect that these drugs may have on the activation of transposable elements. Notably, once reactivated, the GTA-p63 expression could have multifaceted consequences, influencing the regulation of processes such as proliferation, senescence, and apoptosis [54], and also producing new antigens derived from the LTR sequence [22,53]. Our findings show that dual DNMT1 and HDAC inhibition in two glioblastoma cell lines led to a hundreds-fold increase in the expression of the GTA-p63 mRNA variant, while TA-p63 was induced at a comparatively much lower level. This is probably because LTRs can act as a very powerful promoter when inserted at the 5′ region of a coding gene. In fact, they can incorporate several transcription factors binding sites, such as sequence motifs that are recognized by Nuclear Transcription Factor Y (NF-Y), Sp1, and GATA Binding Protein (GATA)-2 [8].

This is in line with the observation that the GTA-p63 transcripts were expressed at fivefold higher levels than conventional TA-p63 when tested under identical conditions from the same vector backbone in H1299 human lung carcinoma cells, also producing more greater protein concentrations [12].

Additionally, we found that the cotreatment with DAC and SB939 induced the expression of p21, a gene regulated by p63, at the mRNA and protein level, both in the U87-MG and T98-G cells. It is already known that p21 expression can increase after inhibition of the HDACs in different cancer models [35,55,56,57], and that the inhibition of DNMT1 and HDACs results in increased p21 expression in lung cancer models [58]. Moreover, pro-apoptotic genes, such as PUMA and NOXA, are among the downstream effectors of p63, which can regulate their expression [12]. Indeed, even if we have not directly demonstrated cause and effect, we were able to see a significant upregulation of PUMA in the U87-MG cells after combining the inhibition of DNMT1 and HDAC compared with control conditions, while NOXA was upregulated both in the U87-MG and T98-G cells. Concordantly, in both cell lines, cotreatment with DAC and SB939 led to PARP1 cleavage, further suggesting an activation of apoptotic processes.

Collectively, this evidence suggested that the inhibition of DNMT1 and HDACs specifically promoted the reactivation of the LTR12C-driven TP63 isoform, and that GTA-p63, besides having a pro-apoptotic role, is also expressed in high quantities compared with the canonical form.

As we analyzed the outcomes of a broad-spectrum treatment, we cannot directly relate the results of our experiments to GTA-p63; however, we were able to assess the presence of a specific and uncommon GTA-p63 isoform, including exon U2. This transcript, which has been described in the human testis [12] as a less abundant additional transcription, is instead expressed in the GBM cell lines, similarly to that containing U1 directly linked to exon 2. Since its main predicted protein differs from the two p63 proteins described above (TA-p63 and GTA-p63) for a new amino terminal of 20 residues, this new protein may represent an interesting aspect to be further explored in the search for specific neo-antigens against glioblastoma.

Examining the potential of targeting LTR12C-induced transcription in therapeutic context, recent genome-wide studies of repetitive elements reported a powerful induction of LTR12C expression in human cells following treatment with epigenetic inhibitors [24], pointing out that there is increasing evidence for the clinical development of antigenic peptides derived from transposable elements. Currently, Decitabine (DAC) and Pracinostat (SB939) have been investigated in the treatment of myelodysplastic syndromes and leukemias [59]. The most frequent adverse effects reported were myelosuppression (attributed to Decatibine) and fatigue (attributed to Pracinostat). Combinatory treatment was associated with high rates of treatment discontinuation due to adverse events, and a more specific therapy may be needed to make this regimen viable in some patients. Indeed, the specific activation of a single LTR12C by CRISPR-mediated activation could reduce the potential side effects from indiscriminate genome activation. LTR12C yields promising features in the action against cancer: first, its strength as a gene promoter can enhance GTA-p63 pro-apoptotic functions by sustaining its transcription; second, by constituting the 5′ end of GTA-p63 transcripts, it can be translated in antigenic epitopes useful for the development of specific vaccines against glioblastoma.

## 5. Conclusions

Epigenetic therapy is increasingly being considered as a viable therapy for different diseases. Here, we showed that the combined inhibition of DNA methyltransferase 1 (DNMT1) and histone deacetylases (HDACs) in the T98-G and U87-MG cell lines demonstrated a significant induction of LTR12C expression, which, in turn, led to the induction of several genes downstream, most significantly GTAp63. The combined treatment of these drugs was found to be synergistic, greatly exceeding the sum of the effects of both drugs individually. This phenomenon is not exclusive to our experiment; the combinatory treatment of DNMT1 and HDAC inhibition has been demonstrated to have a synergistic effect in other studies [60,61].

The potential utility of this treatment could be seen particularly in the dramatic upregulation of GTAp63, an LTR12C-induced isoform of p63, which contributed to the induction of pro-apoptotic pathways in the glioblastoma cell lines. Downstream effectors of p63, such as p21, PUMA, and NOXA, were upregulated following the co-treatment, also suggesting the activation of pro-apoptotic mechanisms, further supported by the cleavage of PARP1, which was observed in both GBM cell lines. We also identified an uncommon GTA-p63 isoform variant that contained an exon U2, which served as a potential source of glioblastoma-specific neoantigens. However, it is important to consider that anti-tumoral pathways were not solely upregulated, and we observed an increase in the expression of genes such as *GBP2* and *GBP5*, which are known to promote tumor progression and growth. Future studies should take these factors into account.

This experiment did have a few limitations. The insights drawn from this study were limited to in vitro glioblastoma cell lines, namely, U87-MG and T98-G. These models are unable to comprehensively reproduce the complex in vivo environment of glioblastomas. Further research should assess whether similar results are reproducible in xenograft models. Furthermore, the global epigenetic modulation presents with the activation of some undesired pathways, including oncogenic ones, such as *GBP2* and *GBP5*, as discussed above. There is a pressing need to develop a more targeted delivery to prevent the induction of undesired pathways and produce a more specific therapeutic effect.

## Figures and Tables

**Figure 1 cells-14-00852-f001:**
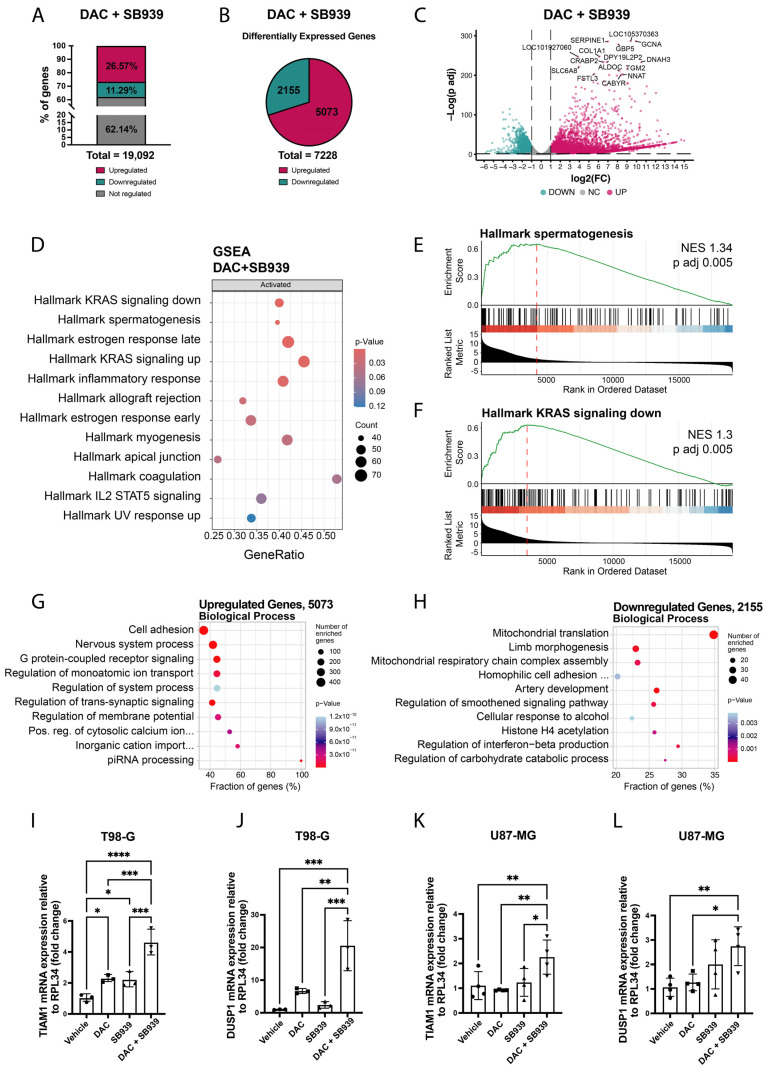
Differential gene expression analysis of the T98-G cells after dual DNMT1 and HDACs inhibition. (**A**) Bar plot of the percentage of genes upregulated, downregulated, or not regulated after the DNMT1 and HDACs inhibition in the T98-G cells (abs FC ≥ 2, *p* adj ≤ 0.05). (**B**) Pie chart of the differentially expressed genes after the combinatory DNMT1 and HDACs inhibition in the T98-G cells with respect to the control (abs FC ≥ 2, *p* adj ≤ 0.05). (**C**) Volcano plot of the differentially expressed genes after the DNMT1 and HDACs inhibition in the T98-G cells with respect to the control. (**D**) Gene Set Enrichment Analysis (GSEA) of the cancer hallmark gene sets from MSigDB in the T98-G cells treated with both DAC and SB939. (**E**,**F**) GSEA of spermatogenesis (**E**) and KRAS signaling down (**F**) signatures in the T98-G cells treated with both DAC and SB939. Bars represent individual genes in the ordered gene set list. The GSEA was performed by implementing the clusterProfiler package in R Studio. (version 2023.06.2+561). (**G**,**H**) Gene Ontology analysis of biological process categories enriched for upregulated (**G**) and downregulated genes (**H**) performed by using the topGO package in the R environment. The *p* value was calculated with the weighted Fisher’s exact test. (**I**,**J**) Quantitative RT-PCR of TIAM1 (**I**) and DUSP1 (**J**) gene expressions after the DNMT1 and/or HDACs inhibition in the T98-G cells. (**K**,**L**) Quantitative RT-PCR of the TIAM1 (**K**) and DUSP1 (**L**) gene expressions after the DNMT1 and/or HDACs inhibition in the U87-MG cells. The results are expressed as $2^{-∆∆Ct}$. The data are expressed as the mean ± SD, * *p* value ≤ 0.05; ** *p* value ≤ 0.01; *** *p* value ≤ 0.001; **** *p* value ≤ 0.0001; test—unpaired two tailed *t*-test of treated samples vs. vehicle, corrected for multiple testing with the Bonferroni method.

**Figure 2 cells-14-00852-f002:**
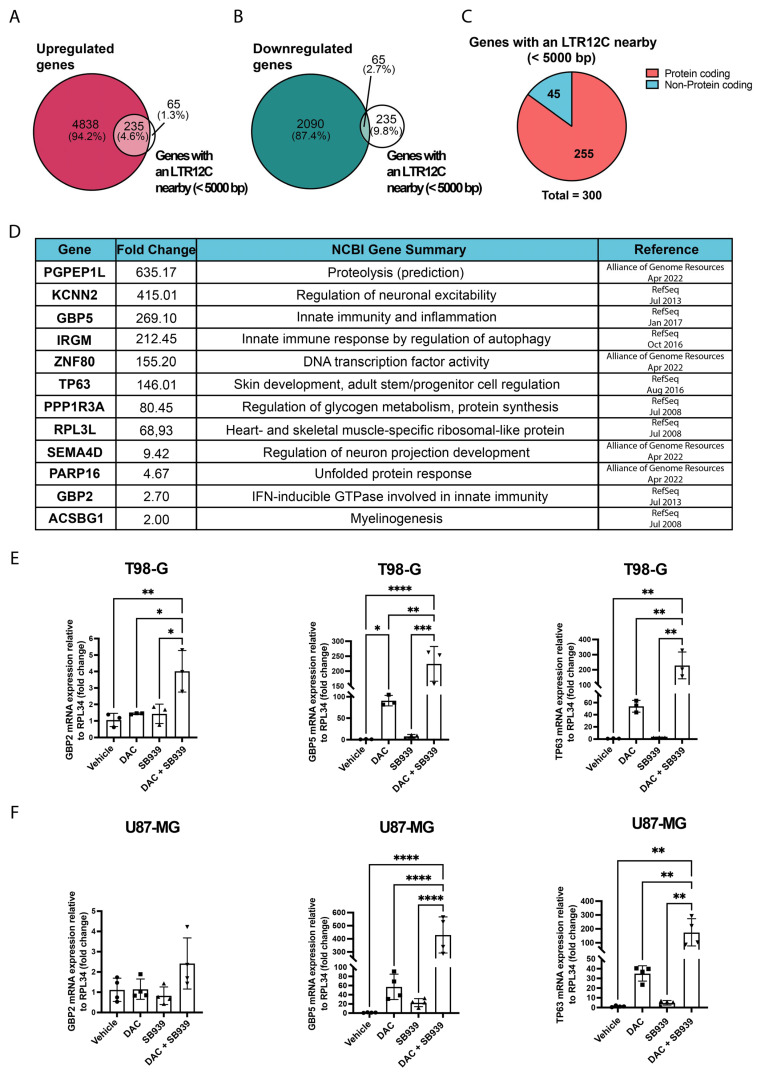
Dual DNMT1 and HDACs inhibition induced the expression of LTR12C-driven genes in the GBM cell models. (**A**) Overlap between the genes upregulated by cotreatment with DAC and SB939 in the T98-G cells and the genes that presented an LTR12C nearby (<5000 bp). (**B**) Overlap between the genes downregulated by the cotreatment with DAC and SB939 in the T98-G cells and the genes that presented a LTR12C nearby (<5000 bp). (**C**) Pie chart of protein coding and non-protein coding genes near to a LTR12C (<5.000 bp) and deregulated by the double inhibition of DNMT1 and HDACs in the T98-G cells. (**D**) Table of genes with an LTR12C that overlapped their TSSs and was deregulated after the cotreatment with DAC and SB939. KCNN2, PPP1R3A, and ACSBG1 were also found on the list provided by Othani et al. (**E**) Quantitative RT-PCR analysis of the GBP2, GBP5, and TP63 mRNA expression levels in the T98-G cells treated with 500 nM DAC and/or 500 nM SB939. (**F**) Quantitative RT-PCR analysis of the GBP2, GBP5, and TP63 global mRNA expressions in the U87-MG cells treated with 500 nM DAC and/or 500 nM SB939. The results are expressed as $2^{-∆∆Ct}$. The data are expressed as the mean ± SD, * *p* value ≤ 0.05; ** *p* value ≤ 0.01; *** *p* value ≤ 0.001; **** *p* value ≤ 0.0001; test—unpaired two tailed *t*-test of treated samples vs. vehicle, corrected for multiple testing with the Bonferroni method.

**Figure 3 cells-14-00852-f003:**
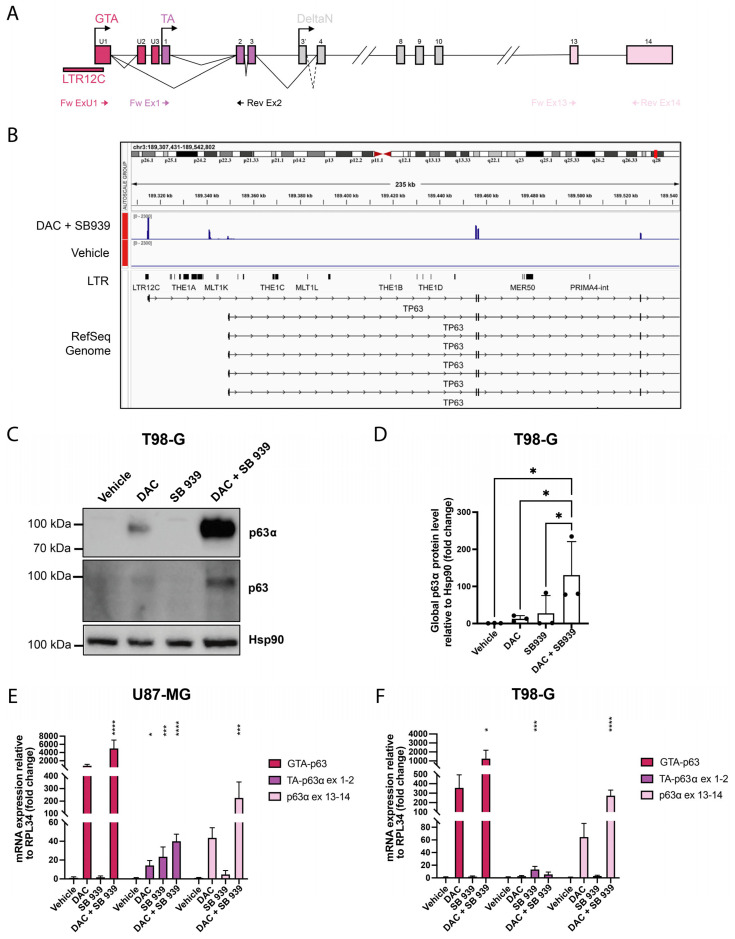
DNMT1 and HDACs combined inhibition induced the GTA-p63 expression at the mRNA and protein levels in glioblastoma cell lines. (**A**) Schematic representation of the TP63 gene locus, showing three different alternative promoters (GTA, TA, and ΔN) and the position of primers used for the RT-qPCR analyses. (**B**) Gene body coverage of the RNA-seq reads on TP63 first exons in the T98-G cells treated with DAC and SB939 or vehicle shows that these drugs induced LTR12C-driven GTA-p63 expression, as visualized on IGV software. (**C**,**D**) Representative Western blot analysis of the p63 protein levels in T98-G after the treatment with 500 nM DAC and/or 500 nM SB939 for 96 h. (**D**) Densitometric analysis of C, performed with ImageJ (1.3K) software. (**E**,**F**) Quantitative RT-PCR analysis of GTA-p63, TA-p63*α*, and global TP63*α* levels in U87-MG (**E**) and T98-G (**F**) cells treated with 500 nM DAC and/or 500 nM SB939 for 96 h. The results are expressed as $2^{-∆∆Ct}$. The data are expressed as the mean ± SD, * *p* value ≤ 0.05; *** *p* value ≤ 0.001; **** *p* value ≤ 0.0001; test—unpaired two tailed *t*-test of treated samples vs. vehicle, corrected for multiple testing with the Bonferroni method.

**Figure 4 cells-14-00852-f004:**
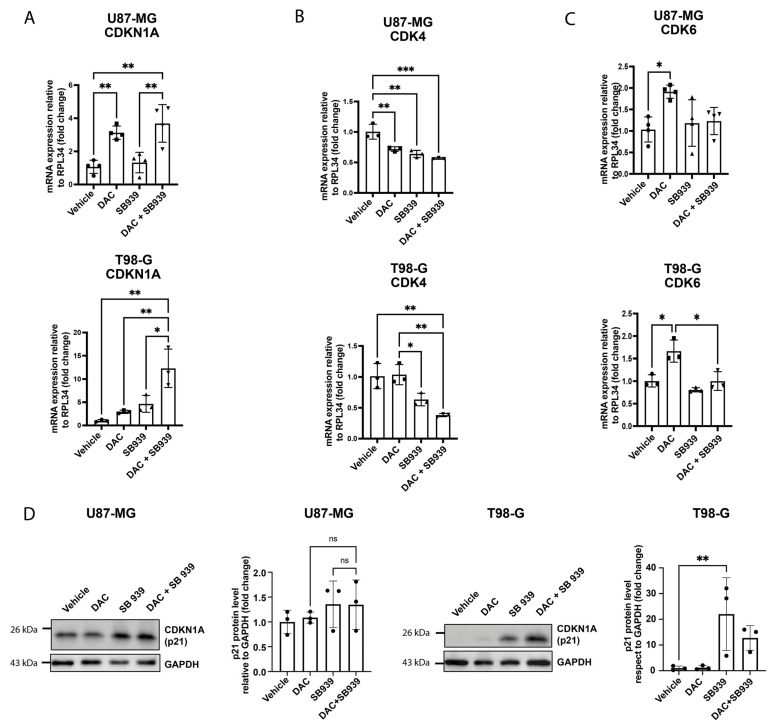
DNMT1 and HDACs combined inhibition affected the expression of the cell cycle-related genes. (**A**) Quantitative RT-PCR of the CDKN1A (p21) mRNA levels after the treatment with 500 nM DAC and/or 500 nM SB939 in the U87-MG (up) and T98-G (down). (**B**) Quantitative RT-PCR of the CDK4 mRNA levels after treatment with 500 nM DAC and/or 500 nM SB939 in the U87-MG (up) and T98-G (down). (**C**) Quantitative RT-PCR of the CDK6 mRNA levels after the treatment with 500 nM DAC and/or 500 nM SB939 in the U87-MG (up) and T98-G (down). The results are expressed as $2^{-∆∆Ct}$. (**D**) Representative Western blot and densitometric analysis of CDKN1A (p21) protein levels in the U87-MG (left) and T98-G (right) cells after the inhibition of DNMT1 and/or HDACs. The data are expressed as the mean ± SD, * *p* value ≤ 0.05; ** *p* value ≤ 0.01; *** *p* value ≤ 0.001; test—unpaired two tailed *t*-test of treated samples vs. vehicle, corrected for multiple testing with the Bonferroni method.

**Figure 5 cells-14-00852-f005:**
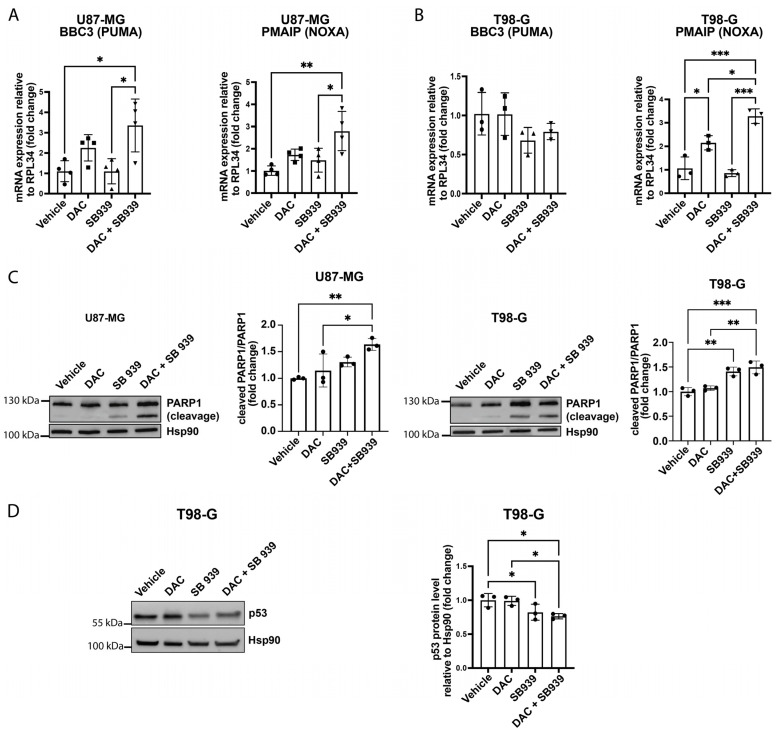
DNMT1 and HDAC combinatory inhibition induced the expression of pro-apoptotic genes. (**A**,**B**) Quantitative RT-PCR of BBC3 (PUMA) and PMAIP (NOXA) mRNA expression levels after the treatment with 500 nM DAC and/or 500 nM SB939 in the U87-MG (**A**) and T98-G (**B**) cells. The results are expressed as $2^{-∆∆Ct}$. (**C**) Representative Western blot and densitometric analysis of the PARP1 protein cleavage after the treatment with 500 nM DAC and/or 500 nM SB939 in the U87-MG (left) and T98-G (right) cells. (**D**) Representative Western blot analysis of the p53 protein expression level after the treatment with 500 nM DAC and/or 500 nM SB939 in the T98-G cells (left) and densitometric analysis (right). The data is expressed as the mean ± SD, * *p* value ≤ 0.05; ** *p* value ≤ 0.01; *** *p* value ≤ 0.001; test—unpaired two tailed *t*-test of treated samples vs. vehicle, corrected for multiple testing with the Bonferroni method.

**Figure 6 cells-14-00852-f006:**
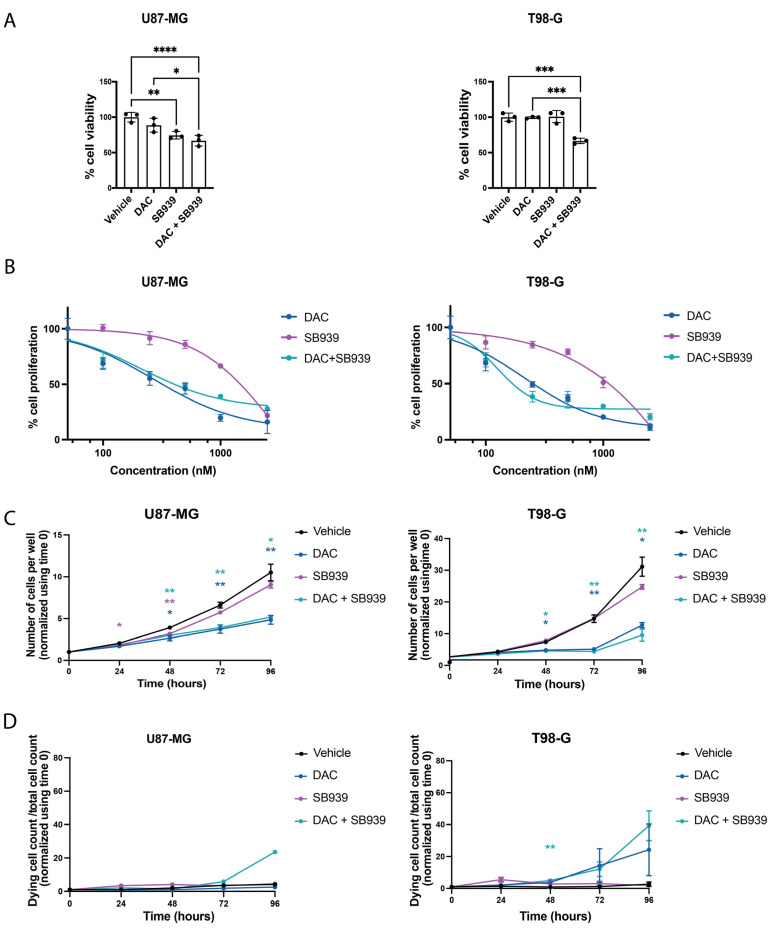
Dual inhibition of DNMT1 and HDACs reduced the cell viability in the two glioblastoma cell models. (**A**) MTS analysis of the U87-MG (left) and T98-G (right) cells treated with 500 nM DAC and/or 500 nM SB939 for 96 h. (**B**) Dose–response curves of DAC, SB939, and DAC and SB939 in the U87-MG (left) and T98-G (right) cells treated for 96 h. The percentage of cell proliferation was calculated from Incucyte Live-Cell Analysis data as the percentage of the ratio of cell number of treated samples over the cell number of the control group. (**C**) Incucyte^®^ Live-Cell Analysis of the U87-MG and T98-G cells during treatment with 500 nM DAC and/or 500 nM SB939 for 96 h. (**D**) Incucyte^®^ Live-Cell Analysis of the numbers of dying U87-MG and T98-G cells during the treatments with 500 nM DAC and/or 500 nM SB939 for 96 h. The data are expressed as the mean ± SD, * *p* value ≤ 0.05; ** *p* value ≤ 0.01; *** *p* value ≤ 0.001; **** *p* value ≤ 0.0001; test—unpaired two tailed *t*-test of treated samples vs. vehicle for each time point, corrected for multiple testing with the Bonferroni method.

**Table 1 cells-14-00852-t001:** IC_50_ of indicated drugs in glioblastoma cell lines.

IC_50_ (nM)	U87-MG	T98-G
**DAC**	332 ± 97	248 ± 35
**SB939**	1327 ± 74	1048 ± 136
**DAC + SB939**	406 ± 133	175 ± 42
**Bliss Score** **(Combination Index)**	−1472	−1082

## Data Availability

All data generated or analyzed during this study are included in this article. Further inquiries can be directed to the senior corresponding author.

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
