# Peer review of "Functional Characterization of LTR12C as Regulators of Germ-Cell-Associated TA-p63 in U87-MG and T98-G In Vitro Models"

_cells, 2025, doi:10.3390/cells14110852_

Round 1

Reviewer 1 Report

Comments and Suggestions for Authors

The current manuscript entitled “Functional characterization of LTR12C as regulators of Germ-cell associated TA-p63 in glioblastoma models” by Meola et al. focuses on the functional role of LTR12C elements in glioblastoma (GBM). The authors demonstrate that pharmacological inhibition of DNA methyltransferases (DNMTs) and histone deacetylases (HDACs) activates the expression of LTR12C and its associated genes in vitro. They further explore the upregulation of the GTA-p63 axis, which appears to exert a pro-apoptotic effect in GBM cell lines. Through a series of biochemical and bioinformatic analyses, the authors provide substantial data to elucidate the regulatory impact of DNMT and HDAC inhibition on LTR12C-driven gene expression in the context of GBM.

Specific Comments:

I recommend authors to perform shRNA- or siRNA-mediated knockdown of HDAC and DNMT to determine the specificity of LTR12C expression in GBM.

Notably, the ICâ‚…â‚€ values of the indicated drugs in U87-MG cells (DAC alone: 320.4 nM; DAC + SB939 combination: 405.4 nM) should be presented with statistical means and standard deviations. I recommend that the authors calculate the Bliss score to better assess the combinatorial effect of DNMT and HDAC inhibitors.

Please double-check the molecular weights in the immunoblots. The band for HSP90 appears to migrate above 100 kDa, which may not be accurate, and GAPDH should be around 36 kDa. These should be corrected or clarified. Similarly, in Figiure-3C, molecular weight of p63 and p63-α appears inappropriate.

Quantitative analysis is required for the immunoblot shown in Figure 4D. There appears to be a discrepancy between the mRNA expression of CDKN1A in Figure 4A and the protein levels in Figure 4D for the same treatment group. The authors should address this inconsistency provide quantification for all immunoblot in study.

Author Response

Please refer to the attachment, thank you.

Reviewer 2 Report

Comments and Suggestions for Authors

In this study, the authors attempt to evaluate the efficacy of DNMT and HDAC in treating glioblastoma. Using DAC and SB939, the authors demonstrate that this combination strategy has significant inhibitory effect on U87-MG and T98-G glioblastoma cell line under in-vitro conditions. While the study is interesting, I have the following suggestions:

  • Line 38 -40: How is glioblastoma linked with transposable element? Is there any evidence that correlates the mortality/case frequency/risk of glioblastoma with transposon elements?
  • Please explain why spermatogenesis is significantly enriched after the treatment. Is this a background process or are there significant number of genes that exclusively regulating spermatogenesis are enriched after the treatment?
    1. If it is a background process, please perform the gene enrichment analysis on spermatogenesis and list the top 20 upregulate and downregulate enriched genes within the process.  
  • Line 358: Crossed line within sentences detected.
  • While the combination treatment of DAC and SB939 on U87-MG and T98-G demonstrated better cell inhibitory effect as compared to monotherapy, it is hard to determine whether the combination effect is addition or synergistic. Please perform a synergy study to clarify the synergistic of drugs applied. The recommended reading for the synergy study is PMID: 22737266 and PMID: 35580060
  • The study only includes findings from the in-vitro study using U87-MG and T98-G. Importantly, there is no in-vivo study included that can justify the application of the drug combination under in-vivo condition. Hence, I would recommend the study title to change from broad “glioblastoma model” to specific “U87-MG and T98-G in vitro model”

In light of these, I would like to recommend a revision for this manuscript, and I look forward to the revised manuscript.

Author Response

(The authors gave the same response as above.)

Reviewer 3 Report

Comments and Suggestions for Authors

This is an important/interesting paper about the potential usefulness of LTR12C or DNMT and HDAC inhibition for the treatment of GBM. I have a few comments about it.

<Major>

#Title

The present title may be okay, but reading the whole of the paper, I could find something else that is potentially more interesting/important; for example, DNMT and HDAC inhibition and its effects on GBM cell lines. P63 is merely one of the factors mediating such effects, isn’t it??? A better title representing the whole of the study is possible, isn’t it?

#Discussion

I can understand that this Discussion kind of reviews each of the results shown by the series of Figures and Tables. Each of the contents/subsections in the Discussion is, by itself, understandable and well-written. But because of the topological separation of the Results from the Discussion, or vice versa, it is rather difficult to understand the Discussion: We (The readers) may have to go back to the Results (figures or tables) to understand the corresponding content/subsection in the Discussion. How about addressing each of the contents/subsections in the Discussion within the space of the Results? In this way, a result as well as what is described related to it in the Discussion may be more understandable, and the Discussion can be more concise and less lengthy.

#Possible applicability of DNMT and HDAC inhibition to humans

I could understand that DNMT and HDAC inhibition is a promising strategy to tackle GBM in the future. But I just wonder what or how its applicability to humans would be like. Is it possible to safely use it for humans? What kinds of chemicals/compounds are plausible as therapeutic agents for DNMT and HDAC inhibition for humans (In this study, they were DAC and SB939, I guess)? What are possible adverse effects? To draw a conclusion is likely impossible, but I think that the comments to such an effect are likely possible by the authors.

<Minor>

#Suppl Fig 1A

Seeing DAC, it shows that DAC was applied three times (24, 48, and 78h). On the other hand, when seeing SB939, it shows that SB939 was applied just one time (78h). I just wonder why the treatment with DAC or SB939 was different between the two. The same condition, i.e., either three times or one time for both, is more reasonable?

#Fig 1

Fig 1A to 1H are about T98-G cells, right? Here, just the results of T98-G cells, but not those of U87-MG, are shown, and I understand this is because the results of T98-G cells are derived from the reassessment of previously published data (GSE209772), right? This is okay, but I just wonder similar results can also be expected for U87-MG cells. If the authors have some data potentially addressing this question, I want them to describe it somewhere in the MS; unless they have, I want them to describe it as it is somewhere in the MS.

#Line 276-7, ‘which reported deregulated transcripts…the TSS [5]: Sorry but I could not understand this sentence.

#Fig 2D

In the text, it says that ‘Comparing our 12 genes with their gene list, we found that KCNN2, PPP1R3A and ACSBG1 belonged to this category’. If this comment is something important, is it better to highlight KCNN2, PPP1R3A and ACSBG1 in Fig 2D in some way or other? Of course, this depends, after all, on the authors.  

#Line 322-323: GBP2 expression was slightly upregulated in T98-G cells -> GBP2 expression was slightly upregulated in T98-G cells (Fig. 2E)

#3.3.2

-line 404: T98 cell line -> T98-G cell line (?)

-FIG -> Fig.

-line 415: may splice in -> may be spliced in (?)

-line 440, ORF: ORF should be full-spelled unless indicated elsewhere in the MS.

#line 594: exapt???

Author Response

(The authors gave the same response as above.)

Round 2

Reviewer 1 Report

Comments and Suggestions for Authors

Authors have addressed most of concern.

Author Response

Thank you for your time.

Reviewer 2 Report

Comments and Suggestions for Authors

The authors have address all the comments and suggestion from the previous review. The manuscript is now ready to be published. I have no further comment.

Author Response

Thank you for your time.

Reviewer 3 Report

Comments and Suggestions for Authors

What/how the authors thought about their study when revising it was reasonable, I understand. 

Just one more point, which is about Response 4: Please describe these comments of the authors somewhere in the MS, unless so far described, so that unfamiliar readers would not wonder the same way as this reviewer actually did. 

Author Response

Thank you for your comment. We have added a brief explanation to the manuscript. It can be found from Line 121 to 125.